# BEVWorld: A Multimodal World Model for Autonomous Driving via Unified BEV Latent Space

## Abstract

World models are receiving increasing attention in autonomous driving for their ability to predict potential future scenarios. In this paper, we present *BEVWorld*, a novel approach that tokenizes multimodal sensor inputs into a unified and compact Bird's Eye View (BEV) latent space for environment modeling. The world model consists of two parts: the multi-modal tokenizer and the latent BEV sequence diffusion model. The multi-modal tokenizer first encodes multi-modality information and the decoder is able to reconstruct the latent BEV tokens into LiDAR and image observations by ray-casting rendering in a self-supervised manner. Then the latent BEV sequence diffusion model predicts future scenarios given action tokens as conditions. Experiments demonstrate the effectiveness of BEVWorld in autonomous driving tasks, showcasing its capability in generating future scenes and benefiting downstream tasks such as perception and motion prediction. Code will be available soon.

## 1 Introduction

Autonomous driving has made significant progress in recent years, but it still faces several challenges. First, training a reliable autonomous driving system requires a large amount of precisely annotated data, which is resource-intensive and time-consuming. Thus, exploring how to utilize unlabeled multimodal sensor data within a self-supervised learning paradigm is crucial. Moreover, a reliable autonomous driving system requires not only the ability to perceive the environment but also a comprehensive understanding of environmental information for decision-making.

We claim that the key to addressing these challenges is to construct a multimodal world model for autonomous driving. By modeling the environment, the world model predicts future states and behaviors, empowering the autonomous agent to make more sophisticated decisions. Recently, some world models have demonstrated their practical significance in autonomous driving Hu et al. (2023); Zhang et al. (2024); Yang et al. (2024b). However, most methods are based on a single modality, which cannot adapt to current multisensor, multimodal autonomous driving systems. Due to the heterogeneous nature of multimodal data, integrating them into a unified generative model and seamlessly adapting to downstream tasks remains an unresolved issue.

In this paper, we introduce BEVWorld, a multimodal world model that transforms diverse multimodal data into a unified bird's-eye-view (BEV) representation and performs action-conditioned future prediction within this unified space. Our BEVWorld consists of two parts: a multimodal tokenizer network and a latent BEV sequence diffusion network.

The core capability of the multimodal tokenizer lies in compressing original multimodal sensor data into a unified BEV latent space. This is achieved by transforming visual information into 3D space and aligning visual semantic information with Lidar geometric information in a self-supervised manner using an auto-encoder structure. To reverse this process and reconstruct the multimodal data, a 3D volume representation is constructed from the BEV latent to predict high-resolution images and point clouds using a ray-based rendering technique Yang et al. (2023).

The Latent BEV Sequence Diffusion network is designed to predict future frames of images and point clouds. With the help of a multimodal tokenizer, this task is made easier, allowing for accurate future

BEV predictions. Specifically, we use a diffusion-based method with a spatial-temporal transformer, which converts sequential noisy BEV latents into clean future BEV predictions based on the action condition.

To summarize, the main contributions of this paper are:

- We introduced a novel multimodal tokenizer that integrates visual semantics and 3D geometry into a unified BEV representation. The quality of the BEV representation is ensured by innovatively applying a rendering-based method to restore multi-sensor data from BEV. The effectiveness of the BEV representation is validated through ablation studies, visualizations, and downstream task experiments.

- We designed a latent diffusion-based world model that enables the synchronous generation of future multi-view images and point clouds. Extensive experiments on the nuScenes and Carla datasets showcase the leading future prediction performance of multimodal data.

## 2 RELATED WORKS

### 2.1 WORLD MODEL

This part mainly reviews the application of world models in the autonomous driving area, focusing on scenario generation as well as the planning and control mechanism. If categorized by the key applications, we divide the sprung-up world model works into two categories. **(1) Driving Scene Generation**. The data collection and annotation for autonomous driving are high-cost and sometimes risky. In contrast, world models find another way to enrich unlimited, varied driving data due to their intrinsic self-supervised learning paradigms. GAIA-1 Hu et al. (2023) adopts multi-modality inputs collected in the real world to generate diverse driving scenarios based on different prompts (e.g., changing weather, scenes, traffic participants, vehicle actions) in an autoregressive prediction manner, which shows its ability of world understanding. ADriver-I Jia et al. (2023) combines the multimodal large language model and a video latent diffusion model to predict future scenes and control signals, which significantly improves the interpretability of decision-making, indicating the feasibility of the world model as a fundamental model. MUVO Bogdoll et al. (2023) integrates LiDAR point clouds beyond videos to predict future driving scenes in the representation of images, point clouds, and 3D occupancy. Further, Copilot4D Zhang et al. (2024) leverages a discrete diffusion model that operates on BEV tokens to perform 3D point cloud forecasting and OccWorld Zheng et al. (2023) adopts a GPT-like generative architecture for 3D semantic occupancy forecast and motion planning. DriveWorld Min et al. (2024) and UniWorld Min et al. (2023) approach the world model as 4D scene understanding task for pre-training for downstream tasks. **(2) Planning and Control**. MILE Hu et al. (2022) is the pioneering work that adopts a model-based imitation learning approach for joint dynamics future environment and driving policy learning in autonomous driving. DriveDreamer Wang et al. (2023a) offers a comprehensive framework to utilize 3D structural information such as HDMap and 3D box to predict future driving videos and driving actions. Beyond the single front view generation, DriveDreamer-2 Zhao et al. (2024) further produces multi-view driving videos based on user descriptions. TrafficBots Zhang et al. (2023) develops a world model for multimodal motion prediction and end-to-end driving, by facilitating action prediction from a BEV perspective. Drive-WM Wang et al. (2023b) generates controllable multiview videos and applies the world model to safe driving planning to determine the optimal trajectory according to the image-based rewards.

### 2.2 VIDEO DIFFUSION MODEL

World model can be regarded as a sequence-data generation task, which belongs to the realm of video prediction. Many early methods Hu et al. (2022; 2023) adopt VAE Kingma & Welling (2013) and auto-regression Chen et al. (2024) to generate future predictions. However, the VAE suffers from unsatisfactory generation quality, and the auto-regressive method has the problem of cumulative error. Thus, many researchers switch to study on diffusion-based future prediction methods Zhao et al. (2024); Li et al. (2023), which achieves success in the realm of video generation recently and has ability to predict multiple future frames simultaneously. This part mainly reviews the related methods of video diffusion model.

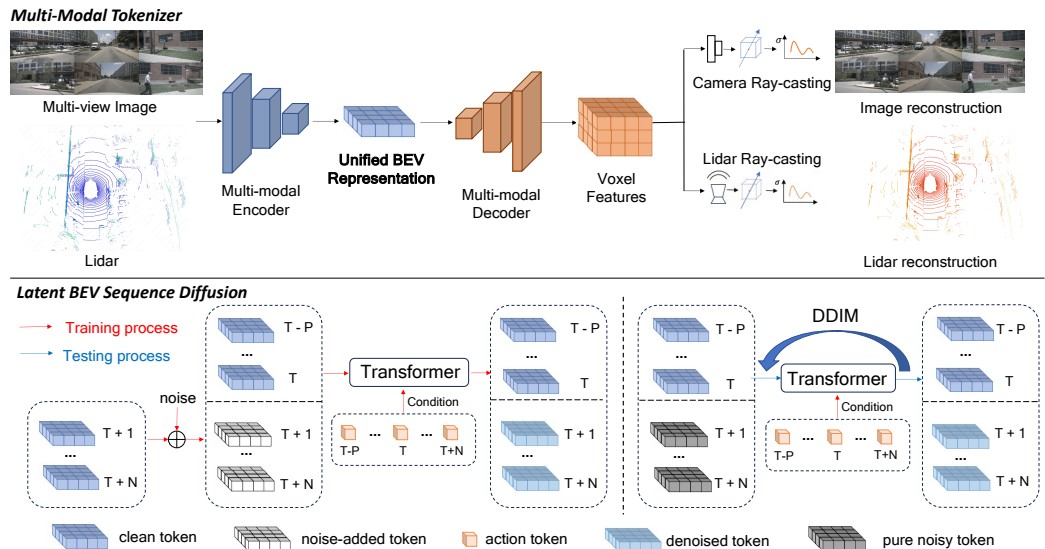

Figure 1: An overview of our method BEVWorld. BEVWorld consists of the multi-modal tokenizer and the latent BEV sequence diffusion model. The tokenizer first encodes the image and Lidar observations into BEV tokens, then decodes the unified BEV tokens to reconstructed observations by NeRF rendering strategies. Latent BEV sequence diffusion model predicts future BEV tokens with corresponding action conditions by a Spatial-Temporal Transformer. The multi-frame future BEV tokens are obtained by a single inference, avoiding the cumulative errors of auto-regressive methods.

The standard video diffusion model Ho et al. (2022) takes temporal noise as input, and adopts the UNet Ronneberger et al. (2015) with temporal attention to obtain denoised videos. However, this method requires high training costs and the generation quality needs further improvement. Subsequent methods are mainly improved along these two directions. In view of the high training cost problem, LVDMHe et al. (2022) and Open-Sora Lab & etc. (2024) methods compress the video into a latent space through schemes such as VAE or VideoGPT Yan et al. (2021), which reduces the video capacity in terms of spatial and temporal dimensions. In order to improve the generation quality of videos, stable video diffusion Blattmann et al. (2023) proposes a multi-stage training strategy, which adopts image and low-resolution video pretraining to accelerate the model convergence and improve generation quality. GenAD Yang et al. (2024a) introduces the causal mask module into UNet to predict plausible futures following the temporal causality. VDT Lu et al. (2023a) and Sora Brooks et al. (2024) replace the traditional UNet with a spatial-temporal transformer structure. The powerful scale-up capability of the transformer enables the model to fit the data better and generates more reasonable videos.

## 3 METHOD

In this section, we delineate the model structure of BEVWorld. The overall architecture is illustrated in Figure 1. Given a sequence of multi-view image and Lidar observations $\{o_{t-P}, \cdots, o_{t-1}, o_t, o_{t+1}, \cdots, o_{t+N}\}$ where $o_t$ is the current observation, $+/-$ represent the future/past observations and $P/N$ is the number of past/future observations, we aim to predict $\{o_{t+1}, \cdots, o_{t+N}\}$ with the condition $\{o_{t-P}, \cdots, o_{t-1}, o_t\}$. In view of the high computing costs of learning a world model in original observation space, a multi-modal tokenizer is proposed to compress the multi-view image and Lidar information into a unified BEV space by frame. The encoder-decoder structure and the self-supervised reconstruction loss promise proper geometric and semantic information is well stored in the BEV representation. This design exactly provides a sufficiently concise representation for the world model and other downstream tasks. Our world model is designed as a diffusion-based network to avoid the problem of error accumulating as those in an auto-regressive fashion. It takes the ego motion and $\{x_{t-P}, \cdots, x_{t-1}, x_t\}$, i.e. the BEV representation of $\{o_{t-P}, \cdots, o_{t-1}, o_t\}$, as condition to learn the noise $\{\epsilon_{t+1}, \cdots, \epsilon_{t+N}\}$ added to $\{x_{t+1}, \cdots, x_{t+N}\}$

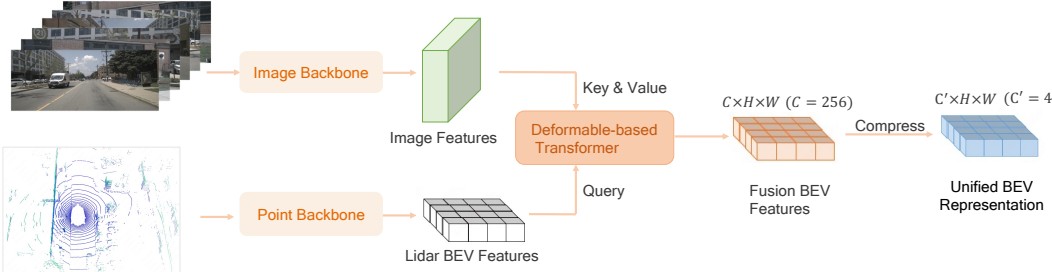

Figure 2: The detailed structure of BEV encoder. The encoder takes as input the multi-view multi-modality sensor data. Multimodal information is fused using deformable attention, BEV features are channel-compressed to be compatible with the diffusion models.

in the training process. In the testing process, a DDIM Song et al. (2020) scheduler is applied to restore the future BEV token from pure noises. Next we use the decoder of multi-modal tokenizer to render future multi-view images and Lidar frames out.

## 3.1 MULTI-MODAL TOKENIZER

Our designed multi-modal tokenizer contains three parts: a BEV encoder network, a BEV Decoder network and a multi-modal rendering network. The structure of BEV encoder network is illustrated in the Figure 2. To make the multi-modal network as homogeneous as possible, we adopt the Swin-Transformer Liu et al. (2021) network as the image backbone to extract multi-image features. For Lidar feature extraction, we first split point cloud into pillars Lang et al. (2019) on the BEV space. Then we use the Swin-Transformer network as the Lidar backbone to extract Lidar BEV features. We fuse the Lidar BEV features and the multi-view images features with a deformable-based transformer Zhu et al. (2020). Specifically, we sample $K(K = 4)$ points in the height dimension of pillars and project these points onto the image to sample corresponding image features. The sampled image features are treated as values and the Lidar BEV features is served as queries in the deformable attention calculation. Considering the future prediction task requires low-dimension inputs, we further compress the fused BEV feature into a low-dimensional($C' = 4$) BEV feature.

For BEV decoder, there is an ambiguity problem when directly using a decoder to restore the images and Lidar since the fused BEV feature lacks height information. To address this problem, we first convert BEV tokens into 3D voxel features through stacked layers of upsampling and swin-blocks. And then we use voxelized NeRF-based ray rendering to restore the multi-view images and Lidar point cloud.

The multi-modal rendering network can be elegantly segmented into two distinct components, image reconstruction network and Lidar reconstruction network. For image reconstruction network, we first get the ray $\mathbf{r}(t) = \mathbf{o} + t\mathbf{d}$, which shooting from the camera center $\mathbf{o}$ to the pixel center in direction $\mathbf{d}$. Then we uniformly sample a set of points $\{(x_i, y_i, z_i)\}_{i=1}^{N_r}$ along the ray, where $N_r(N_r = 150)$ is the total number of points sampled along a ray. Given a sampled point $(x_i, y_i, z_i)$, the corresponding features $\mathbf{v}_i$ are obtained from the voxel feature according to its position. Then, all the sampled features in a ray are aggregated as pixel-wise feature descriptor (Eq. 1).

$$\mathbf{v}(\mathbf{r}) = \sum_{i=1}^{N_r} w_i \mathbf{v_i}, w_i = \alpha_i \prod_{j=1}^{i-1}(1 - \alpha_j), \alpha_i = \sigma(\text{MLP}(\mathbf{v_i})) \tag{1}$$

We traverse all pixels and obtain the 2D feature map $\mathbf{V} \in \mathbb{R}^{H_f \times W_f \times C_f}$ of the image. The 2D feature is converted into the RGB image $\mathbf{I_g} \in \mathbb{R}^{H \times W \times 3}$ through a CNN decoder. Three common losses are added for improving the quality of generated images, perceptual loss Johnson et al. (2016), GAN loss Goodfellow et al. (2020) and L1 loss. Our full objective of image reconstruction is:

$$\mathcal{L}_{\text{rgb}} = \|\mathbf{I_g} - \mathbf{I_t}\|_1 + \lambda_{\text{perc}} \| \sum_{j=1}^{N_\phi} \phi^j(\mathbf{I_g}) - \phi^j(\mathbf{I_t})\| + \lambda_{\text{gan}}\mathcal{L}_{\text{gan}}(\mathbf{I_g}, \mathbf{I_t}) \tag{2}$$

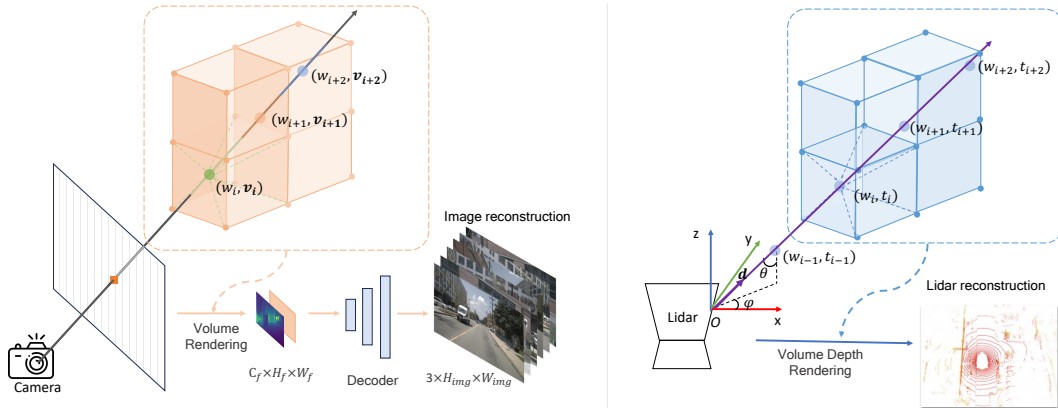

Figure 3: **Left**: Details of the multi-view images rendering. Trilinear interpolation is applied to the series of sampled points along the ray to obtain weight $w_i$ and feature $\mathbf{v}_i$. $\{\mathbf{v}_i\}$ are weighted by $\{w_i\}$ and summed, respectively, to get the rendered image features, which are concatenated and fed into the decoder for $8\times$ upsampling, resulting in multi-view RGB images. **Right**: Details of Lidar rendering. Trilinear interpolation is also applied to obtain weight $w_i$ and depth $t_i$. $\{t_i\}$ are weighted by $\{w_i\}$ and summed, respectively, to get the final depth of point. Then the point in spherical coordinate system is transformed to the Cartesian coordinate system to get vanilla Lidar point coordinate.

where $\mathbf{I_t}$ is the ground truth of $\mathbf{I_g}$, $\phi^j$ represents the jth layer of pretrained VGG Simonyan & Zisserman (2014) model, and the definition of $\mathcal{L}_{\text{gan}}(\mathbf{I_g}, \mathbf{I_t})$ can be found in Goodfellow et al. (2020).

For Lidar reconstruction network, the ray is defined in the spherical coordinate system with inclination $\theta$ and azimuth $\phi$. $\theta$ and $\phi$ are obtained by shooting from the Lidar center to current frame of Lidar point. We sample the points and get the corresponding features in the same way of image reconstruction. Since Lidar encodes the depth information, the expected depth $D_g(\mathbf{r})$ of the sampled points are calculated for Lidar simulation. The depth simulation process and loss function are shown in Eq. 3.

$$D_g(\mathbf{r}) = \sum_{i=1}^{N_r} w_i t_i, \quad \mathcal{L}_{\text{Lidar}} = \|D_g(\mathbf{r}) - D_t(\mathbf{r})\|_1, \tag{3}$$

where $t_i$ denotes the depth of sampled point from the Lidar center and $D_t(\mathbf{r})$ is the depth ground truth calculated by the Lidar observation.

The Cartesian coordinate of point cloud could be calculated by:

$$(x, y, z) = (D_g(\mathbf{r}) \sin\theta \cos\phi, D_g(\mathbf{r}) \sin\theta \sin\phi, D_g(\mathbf{r}) \cos\theta) \tag{4}$$

Overall, the multi-modal tokenizer is trained end-to-end with the total loss in Eq. 5:

$$\mathcal{L}_{\text{Total}} = \mathcal{L}_{\text{Lidar}} + \mathcal{L}_{\text{rgb}} \tag{5}$$

## 3.2 LATENT BEV SEQUENCE DIFFUSION

Most existing world models Zhang et al. (2024); Hu et al. (2023) adopt autoregression strategy to get longer future predictions, but this method is easily affected by cumulative errors. Instead, we propose latent sequence diffusion framework, which inputs multiple frames of noise BEV tokens and obtains all future BEV tokens simultaneously.

The structure of latent sequence diffusion is illustrated in Figure 1. In the training process, the low-dimensional BEV tokens $(x_{t-P}, \cdots, x_{t-1}, x_t, x_{t+1}, \cdots, x_{t+N})$ are firstly obtained from the sensor data. Only BEV encoder in the multi-modal tokenizer is involved in this process and the parameters of multi-modal tokenizer is frozen. To facilitate the learning of BEV token features by the world model module, we standardize the input BEV features along the channel dimension $(\overline{x}_{t-P}, \cdots, \overline{x}_{t-1}, \overline{x}_t, \overline{x}_{t+1}, \cdots, \overline{x}_{t+N})$. Latest history BEV token and current frame BEV token

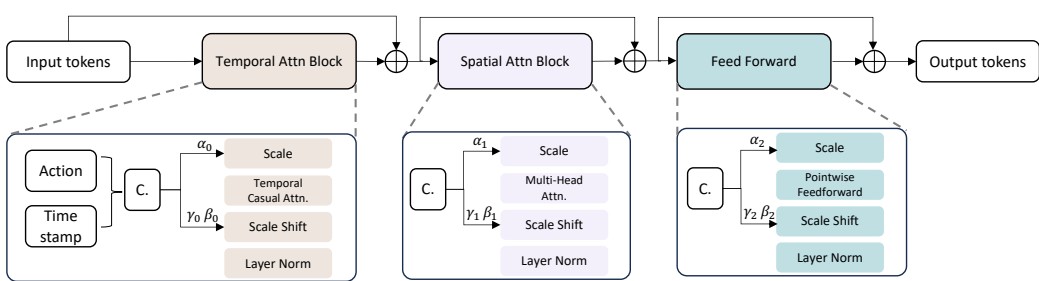

Figure 4: The architecture of Spatial-Temporal transformer block.

$(\overline{x}_{t-P}, \cdots, \overline{x}_{t-1}, \overline{x}_t)$ are served as condition tokens while $(\overline{x}_{t+1}, \cdots, \overline{x}_{t+N})$ are diffused to noisy BEV tokens $(\overline{x}_{t+1}^\epsilon, \cdots, \overline{x}_{t+N}^\epsilon)$ with noise $\{\epsilon_{\hat{t}}^i\}_{i=t+1}^{t+N}$, where $\hat{t}$ is the timestamp of diffusion process.

The denoising process is carried out with a spatial-temporal transformer containing a sequence of transformer blocks, the architecture of which is shown in the Figure 4. The input of spatial-temporal transformer is the concatenation of condition BEV tokens and noisy BEV tokens $(\overline{x}_{t-P}, \cdots, \overline{x}_{t-1}, \overline{x}_t, \overline{x}_{t+1}^\epsilon, \cdots, \overline{x}_{t+N}^\epsilon)$. These tokens are modulated with action tokens $\{a_i\}_{i=T-P}^{T+N}$ of vehicle movement and steering, which together form the inputs to spatial-temporal transformer. More specifically, the input tokens are first passed to temporal attention block for enhancing temporal smoothness. To avoid time confusion problem, we added the causal mask into temporal attention. Then, the output of temporal attention block are sent to spatial attention block for accurate details. The design of spatial attention block follows standard transformer block criterion Lu et al. (2023a). Action token and diffusion timestamp $\{\hat{t}_i^d\}_{i=T-P}^{T+N}$ are concatenated as the condition $\{c_i\}_{i=T-P}^{T+N}$ of diffusion models and then sent to AdaLN Peebles & Xie (2023) (6) to modulate the token features.

$$\mathbf{c} = \text{concat}(\mathbf{a}, \hat{\mathbf{t}}); \; \gamma, \beta = \text{Linear}(\mathbf{c}); \; \text{AdaLN}(\hat{\mathbf{x}}, \gamma, \beta) = \text{LayerNorm}(\hat{\mathbf{x}}) \cdot (1 + \gamma) + \beta \quad (6)$$

where $\hat{\mathbf{x}}$ is the input features of one transformer block, $\gamma, \beta$ is the scale and shift of $\mathbf{c}$.

The output of the Spatial-Temporal transformer is the noise prediction $\{\epsilon_{\hat{t}}^i(\mathbf{x})\}_{i=1}^N$, and the loss is shown in Eq. 7.

$$\mathcal{L}_{\text{diff}} = \|\epsilon_{\hat{\mathbf{t}}}(\mathbf{x}) - \epsilon_{\hat{\mathbf{t}}}\|_1. \quad (7)$$

In the testing process, normalized history frame and current frame BEV tokens $(\overline{x}_{t-P}, \cdots, \overline{x}_{t-1}, \overline{x}_t)$ and pure noisy tokens $(\epsilon_{t+1}, \epsilon_{t+2}, \cdots, \epsilon_{t+N})$ are concatenated as input to world model. The ego motion token $\{a_i\}_{i=T-P}^{T+N}$, spanning from moment $T-P$ to $T+N$, serve as the conditional inputs. We employ the DDIM Song et al. (2020) schedule to forecast the subsequent BEV tokens. Subsequently, the denormalized operation is applied to the predicted BEV tokens, which are then fed into the BEV decoder and rendering network yielding a comprehensive set of predicted multi-sensor data.

## 4 EXPERIMENTS

### 4.1 DATASET

**NuScenes** Caesar et al. (2020) NuScenes is a widely used autonomous driving dataset, which comprises multi-modal data such as multi-view images from 6 cameras and Lidar scans. It includes a total of 700 training videos and 150 validation videos. Each video includes 20 seconds at a frame rate of 12Hz.

**Carla** Dosovitskiy et al. (2017) The training data is collected in the open-source CARLA simulator at 2Hz, including 8 towns and 14 kinds of weather. We collect 3M frames with four cameras (1600 × 900) and one Lidar (32p) for training, and evaluate on the Carla Town05 benchmark, which is the same setting of Shao et al. (2022).

## 4.2 Multi-modal Tokenizer

In this section, we explore the impact of different design decisions in the proposed multi-modal tokenizer and demonstrate its effectiveness in the downstream tasks. For multi-modal reconstruction visualization results, please refer to Figure7 and Figure8.

### 4.2.1 Ablation Studies

**Various input modalities and output modalities.** The proposed multi-modal tokenizer supports various choice of input and output modalities. We test the influence of different modalities, and the results are shown in Table 1, where L indicates Lidar modality, C indicates multi-view cameras modality, and L&C indicates multi-modal modalities. The combination of Lidar and cameras achieves the best reconstruction performance, which demonstrates that using multi modalities can generate better BEV features. We find that the PSNR metric is somewhat distorted when comparing ground truth images and predicted images. This is caused by the mean characteristics of PSNR metric, which does not evaluate sharpening and blurring well. As shown in Figure 12, though the PSNR of multi modalities is slightly lower than single camera modality method, the visualization of multi modalities is better than single camera modality as the FID metric indicates.

Table 1: Ablations of different modalities.

| Input | Output | FID↓ | PSNR↑ | Chamfer↓ |
|-------|--------|-------|--------|----------|
| C | C | 19.18 | **26.95** | - |
| C | L | - | - | 2.67 |
| L | L | - | - | 0.19 |
| L & C | L & C | **5.54** | 25.68 | **0.15** |

Table 2: Ablations of rendering methods.

| Method | FID↓ | PSNR↑ | Chamfer↓ |
|--------|-------|--------|----------|
| (a) | 67.28 | 9.45 | 0.24 |
| (b) | **5.54** | **25.68** | **0.15** |

**Rendering approaches.** To convert from BEV features into multiple sensor data, the main challenge lies in the varying positions and orientations of different sensors, as well as the differences in imaging (points and pixels). We compared two types of rendering methods: a) attention-based method, which implicitly encodes the geometric projection in the model parameters via global attention mechanism; b) ray-based sampling method, which explicitly utilizes the sensor's pose information and imaging geometry. The results of the methods (a) and (b) are presented in Table 2. Method (a) faces with a significant performance drop in multi-view reconstruction, indicating that our ray-based sampling approach reduces the difficulty of view transformation, making it easier to achieve training convergence. Thus we adopt ray-based sampling method for generating multiple sensor data.

### 4.2.2 Benefit for Downstream Tasks

**3D Detection.** To verify our proposed method is effective for downstream tasks when used in the pre-train stage, we conduct experiments on the nuScenes 3D detection benchmark. For the model structure, in order to maximize the reuse of the structure of our multi-modal tokenizer, the encoder in the downstream 3D detection task is kept the same with the encoder of the tokenizer described in 3. We use a BEV encoder attached to the tokenizer encoder for further extracting BEV features. We design a UNet-style network with the Swin-transformer Liu et al. (2021) layers as the BEV encoder. As for the detection head, we adopt query-based head Li et al. (2022), which contains 500 object queries that searching the whole BEV feature space and uses hungarian algorithm to match the prediction boxes and the ground truth boxes. We report both single frame and two frames results. We warp history 0.5s BEV future to current frame in two frames setting for better velocity estimation. Note that we do not perform fine-tuning specifically for the detection task all in the interest of preserving the simplicity and clarity of our setup. For example, the regular detection range is [-60.0m, -60.0m, -5.0m, 60.0m, 60.0m, 3.0m] in the nuScenes dataset while we follow the BEV range of [-80.0m, -80.0m, -4.5m, 80.0m, 80.0m, 4.5m] in the multi-modal reconstruction task, which would result in coarser BEV grids and lower accuracy. Meanwhile, our experimental design eschew the use of data augmentation techniques and the layering of point cloud frames. We train 30 epoches on 8 A100 GPUs with a starting learning rate of $5e^{-4}$ that decayed with cosine annealing policy. We mainly focus on the relative performance gap between training from scratch and use our proposed self-supervised tokenizer as pre-training model. As demonstrated in Table 3, it is evident that employing our multi-modal tokenizer as a pre-training model yields significantly enhanced performance across

both single and multi-frame scenarios. Specifically, with a two-frame configuration, we have achieved an impressive $8.4\%$ improvement in the NDS metric and a substantial $13.4\%$ improvement in the mAP metric, attributable to our multi-modal tokenizer pre-training approach.

**Motion Prediction.** We further validate the performance of using our method as pre-training model on the motion prediction task. We attach the motion prediction head to the 3D detection head. The motion prediction head is stacked of 6 layers of cross attention(CA) and feed-forward network(FFN). For the first layer, the trajectory queries is initialized from the top 200 highest score object queries selected from the 3D detection head. Then for each layer, the trajectory queries is firstly interacting with temporal BEV future in CA and further updated by FFN. We reuse the hungarian matching results in 3D detection head to pair the prediction and ground truth for trajectories. We predict five possible modes of trajectories and select the one closest to the ground truth for evaluation. For the training strategy, we train 24 epoches on 8 A100 GPUs with a starting learning rate of $1e^{-4}$. Other settings are kept the same with the detection configuration. We display the motion prediction results in Table 3. We observed a decrease of 0.455 meters in minADE and a reduction of 0.749 meters in minFDE at the two-frames setting when utilizing the tokenizer during the pre-training phase. This finding confirms the efficacy of self-supervised multi-modal tokenizer pre-training.

Table 3: Comparison of whether use pretrained tokenizer on the nuScenes validation set.

| Frames | Pretrain | 3D Object Detection | | | | | | | Motion Prediction | |
|---|---|---|---|---|---|---|---|---|---|---|
| | | NDS↑ | mAP↑ | mATE↓ | mASE↓ | mAOE↓ | mAVE↓ | mAAE↓ | minADE↓ | minFDE↓ |
| Single | wo | 0.366 | 0.338 | 0.555 | 0.290 | 0.832 | 1.290 | **0.357** | 2.055 | 3.469 |
| Single | w | **0.415** | **0.412** | **0.497** | **0.278** | **0.769** | **1.275** | 0.367 | **1.851** | **3.153** |
| Two | wo | 0.392 | 0.253 | 0.567 | 0.308 | 0.650 | 0.610 | **0.212** | 1.426 | 2.230 |
| Two | w | **0.476** | **0.387** | **0.507** | **0.287** | **0.632** | **0.502** | 0.246 | **0.971** | **1.481** |

Table 4: Comparison of generation quality on nuScenes validation dataset.

| Methods | Multi-view | Video | Manual Labeling Cond. | FID↓ | FVD↓ |
|---|---|---|---|---|---|
| DriveDreamer Wang et al. (2023a) | | ✓ | ✓ | 52.6 | 452.0 |
| WoVoGen Lu et al. (2023b) | ✓ | ✓ | ✓ | 27.6 | 417.7 |
| Drive-WM Wang et al. (2023b) | ✓ | ✓ | ✓ | 15.8 | 122.7 |
| DriveGAN Kim et al. (2021) | | ✓ | | 73.4 | 502.3 |
| Drive-WM Wang et al. (2023b) | ✓ | ✓ | | 20.3 | 212.5 |
| BEVWorld | ✓ | ✓ | | 19.0 | 154.0 |

Table 5: Comparison with SOTA methods on the nuScenes validation set and Carla dataset. The suffix * represents the methods adopt classifier-free guidance (CFG) when getting the final results, and † is the reproduced result. Cham. is the abbreviation of Chamfer Distance.

| Dataset | Methods | Modal | PSNR 1s↑ | FID 1s↓ | Cham. 1s↓ | PSNR 3s↑ | FID 3s↓ | Cham. 3s↓ |
|---|---|---|---|---|---|---|---|---|
| nuScenes | SPFNet Weng et al. (2021) | Lidar | - | - | 2.24 | - | - | 2.50 |
| nuScenes | S2Net Weng et al. (2022) | Lidar | - | - | 1.70 | - | - | 2.06 |
| nuScenes | 4D-Occ Khurana et al. (2023) | Lidar | - | - | 1.41 | - | - | 1.40 |
| nuScenes | Copilot4D* Zhang et al. (2024) | Lidar | - | - | 0.36 | - | - | 0.58 |
| nuScenes | Copilot4D Zhang et al. (2024) | Lidar | - | - | - | - | - | 1.40 |
| nuScenes | BEVWorld | Multi | 20.85 | 22.85 | 0.44 | 19.67 | 37.37 | 0.73 |
| Carla | 4D-Occ† Khurana et al. (2023) | Lidar | - | - | 0.27 | - | - | 0.44 |
| Carla | BEVWorld | Multi | 20.71 | 36.80 | 0.07 | 19.12 | 43.12 | 0.17 |

## 4.3 LATENT BEV SEQUENCE DIFFUSION

In this section, we introduce the training details of latent BEV Sequence diffusion and compare this method with other related methods.

### 4.3.1 TRAINING DETAILS.

**NuScenes.** We adopt a three stage training for future BEV prediction. 1) Next BEV pretraining. The model predicts the next frame with the $\{x_{t-1}, x_t\}$ condition. In practice, we adopt sweep data of

nuScenes to reduce the difficulty of temporal feature learning. The model is trained 20000 iters with a batch size 128. 2) Short Sequence training. The model predicts the $N(N = 5)$ future frames of sweep data. At this stage, the network can learn how to perform short-term (0.5s) feature reasoning. The model is trained 20000 iters with a batch size 128. 3) Long Sequence Fine-tuning. The model predicts the $N(N = 6)$ future frames (3s) of key-frame data with the $\{x_{t-2}, x_{t-1}, x_t\}$ condition. The model is trained 30000 iters with a batch size 128. The learning rate of three stages is 5e-4 and the optimizer is AdamW Loshchilov & Hutter (2017). Note that our method does not introduce classifier-free guidance (CFG) strategy in the training process for better integration with downstream tasks, as CFG requires an additional network inference, which doubles the computational cost.

**Carla.** The model is fine-tuned 30000 iterations with a nuScenes-pretrained model with a batch size 32. The initial learning rate is 5e-4 and the optimizer is AdamW Loshchilov & Hutter (2017). CFG strategy is not introduced in the training process, following the same setting of nuScenes.

### 4.3.2 LIDAR PREDICTION QUALITY

**NuScenes.** We compare the Lidar prediction quality with existing SOTA methods. We follow the evaluation process of Zhang et al. (2024) and report the Chamfer 1s/3s results in Table 5, where the metric is computed within the region of interest: -70m to +70m in both x-axis and y-axis, -4.5m to +4.5m in z-axis. Our proposed method outperforms SPFNet, S2Net and 4D-Occ in Chamfer metric by a large margin. When compared to Copilot4D Zhang et al. (2024), our approach uses less history condition frames and no CFG schedule setting considering the large memory cost for multi-modal inputs. Our BEVWorld requires only 3 past frames for 3-second predictions, whereas Copilot4D utilizes 6 frames for the same duration. Our method demonstrates superior performance, achieving chamfer distance of 0.73 compared to 1.40, in the no CFG schedule setting, ensuring a fair and comparable evaluation.

**Carla**. We also conducted experiments on the Carla dataset to verify the scalability of our method. The quantitative results are shown in Table 5. We reproduce the results of 4D-Occ on Carla and compare it with our method, obtaining similar conclusions to this on the nuScenes dataset. Our method significantly outperform 4D-Occ in prediction results for both 1-second and 3-second.

### 4.3.3 VIDEO GENERATION QUALITY

**NuScenes.** We compare the video generation quality with past single-view and multi-view generation methods. Most of existing methods adopt manual labeling condition, such as layout or object label, to improve the generation quality. However, using annotations reduces the scalability of the world model, making it difficult to train with large amounts of unlabeled data. Thus we do not use the manual annotations as model conditions. The results are shown in Table 4. The proposed method achieves best FID and FVD performance in methods without using manual labeling condition and exhibits comparable results with methods using extra conditions. The visual results of Lidar and video prediction are shown in Figure 5. Furthermore, the generation can be controlled by the action conditions. We transform the action token into left turn, right turn, speed up and slow down, and the generated image and Lidar can be generated according to these instructions. The visualization of controllability are shown in Figure 6.

**Carla.** The generation quality on Carla is similar to that on nuScenes dataset, which demonstrates the scalability of our method across different datasets. The quantitative results of video predictions are shown in Table 4 with 36.80(FID 1s) and 43.12(FID 3s). Qualitative results of video predictions are shown in the appendix.

### 4.3.4 BENEFIT FOR PLANNING TASKS

We further validate the effectiveness of the predicted future BEV features from latent diffusion network for toy downstream open-loop planning task Zhai et al. (2023) on nuScenes dataset. Note that we do not use actions of ego car in future frames here and we adopt $x_0$-parameterization Austin et al. (2021) for fast inference. We adopt four vectors, history trajectory, command, perception and optional future BEV vectors, as input for planning head. History trajectory vector encodes the ego movement from last frame to current frame. Command vector refers to the routing command such as turning left or right. Perception vector is extracted from the object query in the detection head that interacted with all detection queries. Future BEV vector is obtained from the pooled BEV features

from the fixed diffusion model. When using future BEV vectors, PNC L2 3s metric is decreased from **1.030m** to **0.977m**, which validates that the predicted BEV from world model is beneficial for planning tasks.

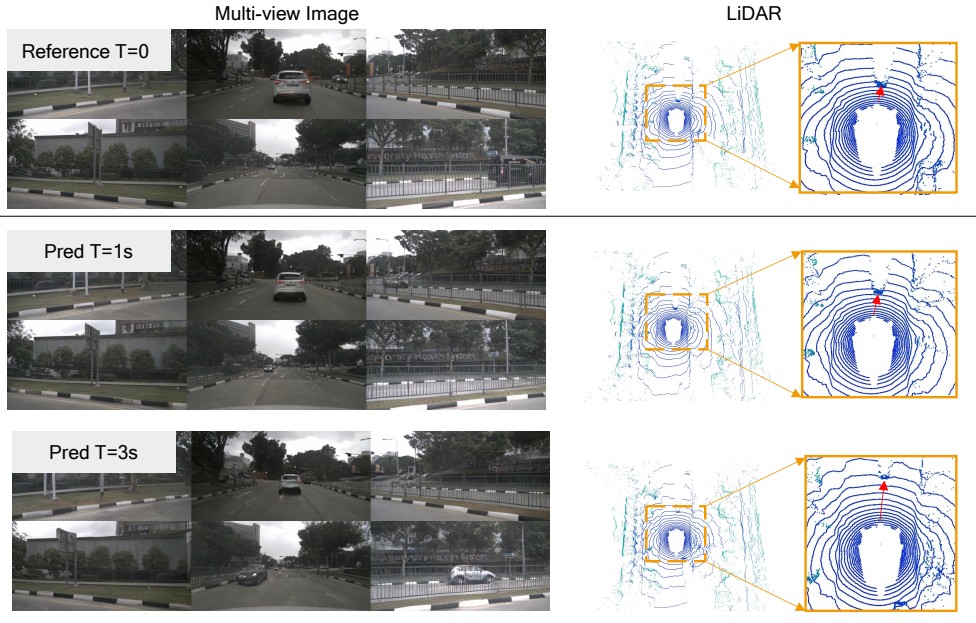

Figure 5: The visualization of Lidar and video predictions.

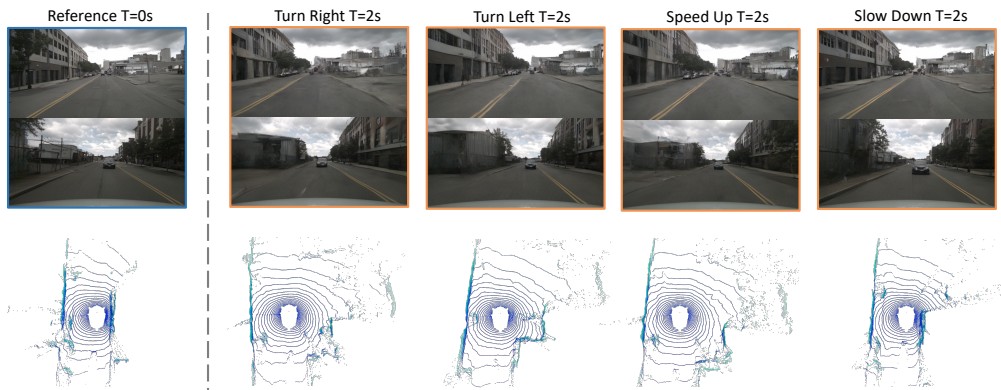

Figure 6: The visualization of controllability. Due to space limitations, we only show the results of the front and rear views for a clearer presentation.

## 5 CONCLUSION

We present BEVWorld, an innovative autonomous driving framework that leverages a unified Bird's Eye View latent space to construct a multi-modal world model. BEVWorld's self-supervised learning paradigm allows it to efficiently process extensive unlabeled multimodal sensor data, leading to a holistic comprehension of the driving environment. We validate the effectiveness of BEVWorld in the downstream autonomous driving tasks. Furthermore, BEVWorld achieves satisfactory results in multi-modal future predictions with latent diffusion network, showcasing its capabilities through experiments on both real-world(nuScenes) and simulated(carla) datasets. We hope that the work presented in this paper will stimulate and foster future developments in the domain of world models for autonomous driving.

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

APPENDIX

# A QUALITATIVE RESULTS

In this section, qualitative results are presented to demonstrate the performance of the proposed method.

## A.1 TOKENIZER RECONSTRUCTIONS

The visualization of tokenizer reconstructions are shown in Figure 7 and Figure 8. The proposed tokenizer can recover the image and Lidar with the unified BEV features.

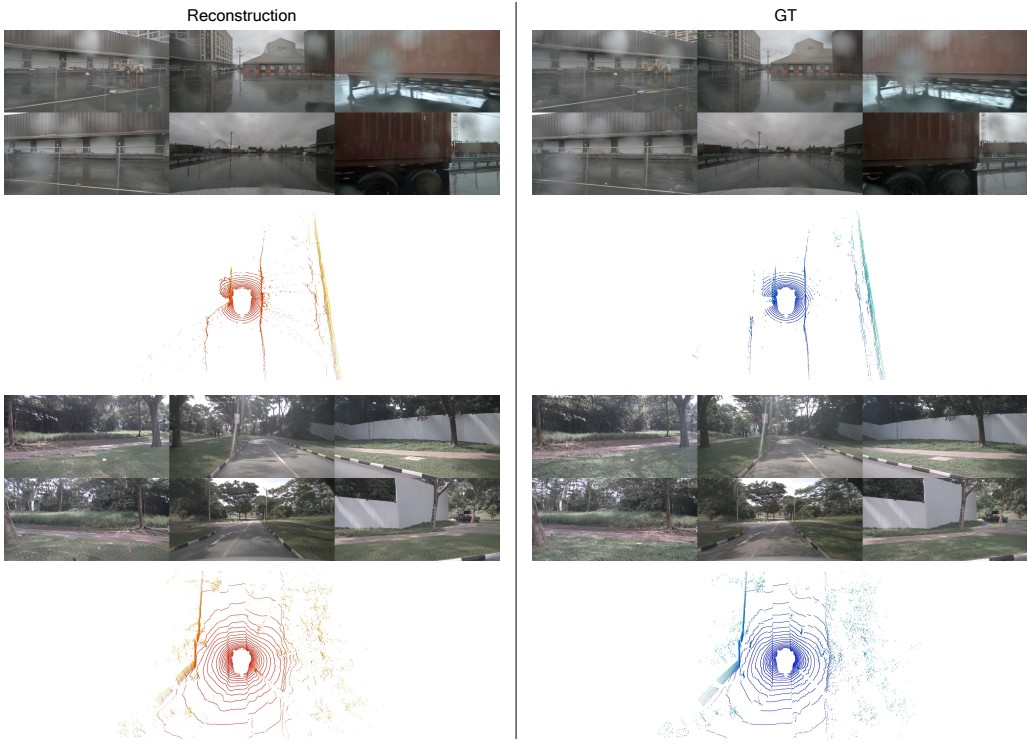

Figure 7: The visualization of LiDAR and video reconstructions on nuScenes dataset.

## A.2 MULTI-MODAL FUTURE PREDICTIONS

**Diverse generation.** The proposed diffusion-based world model can produce high-quality future predictions with different driving conditions, and both the dynamic and static objects can be generated properly. The qualitative results are illustrated in Figure 9 and Figure 10.

**Controllability.** We present more visual results of controllability in Figure 11. The generated images and Lidar exhibit a high degree of consistency with action, which demonstrates that our world model has the potential of being a simulator.

**PSNR metric.** PSNR metric has the problem of being unable to differentiate between blurring and sharpening. As shown in Figure 12, the image quality of L & C is better the that of C, while the psnr metric of L & C is worse than that of C.

# B IMPLEMENTATION DETAILS

**Training details of tokenizer.** We trained our model using 32 GPUs, with a batch size of 1 per card. We used the AdamW optimizer with a learning rate of 5e-4, beta1=0.5, and beta2=0.9, following a

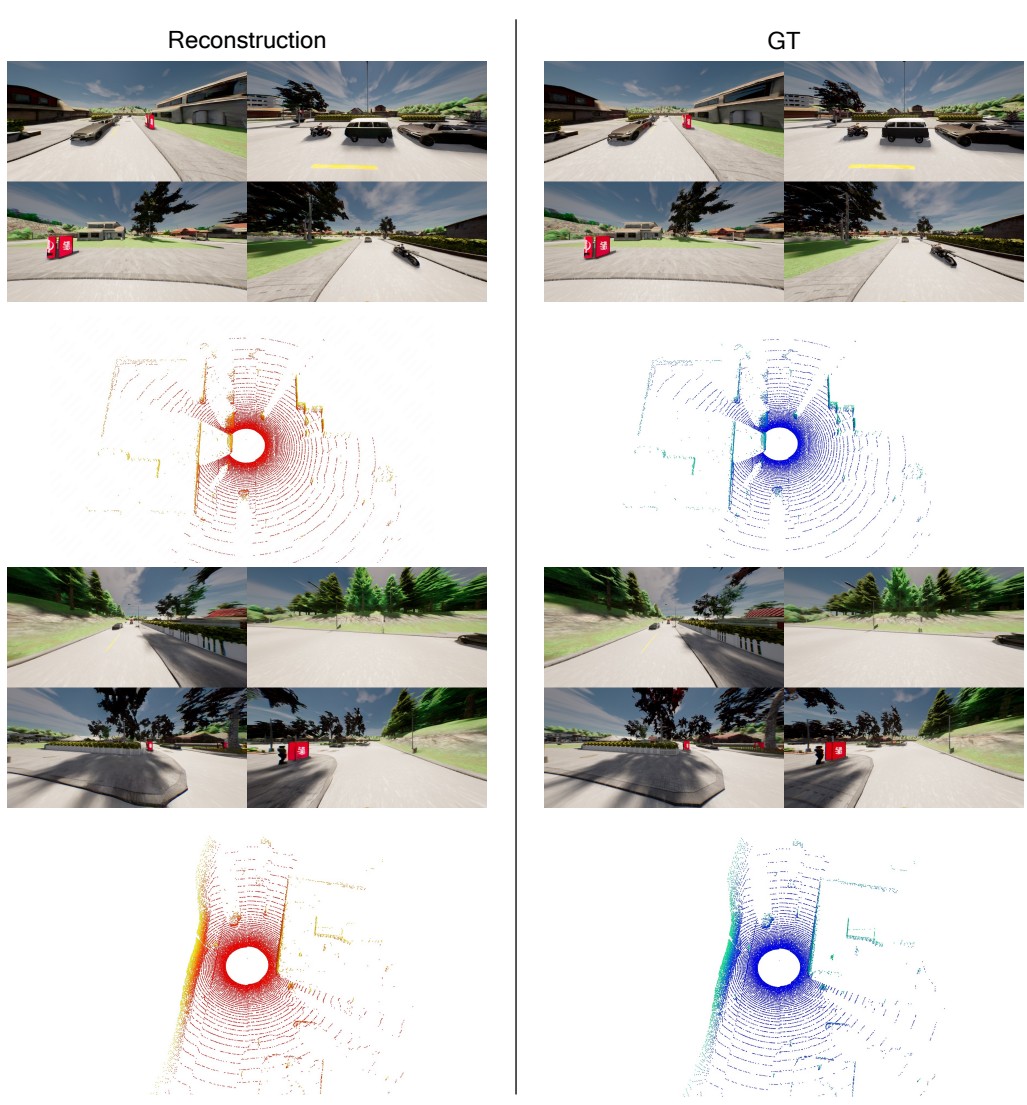

Figure 8: The visualization of LiDAR and video reconstructions on Carla dataset.

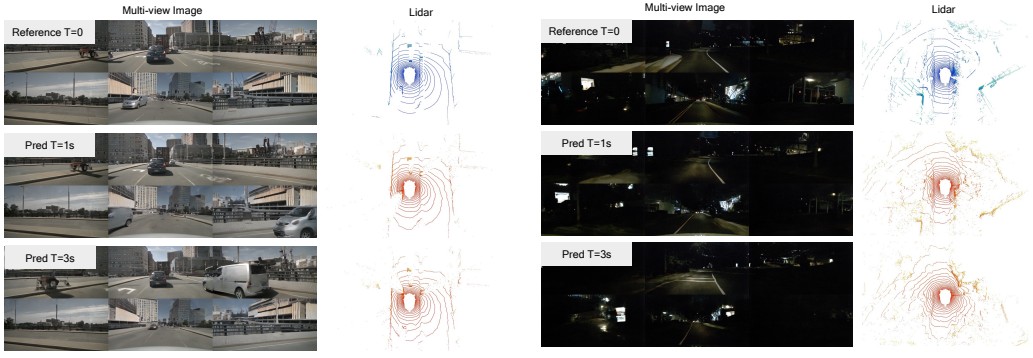

Figure 9: The visualization of LiDAR and future predictions on nuScenes dataset.

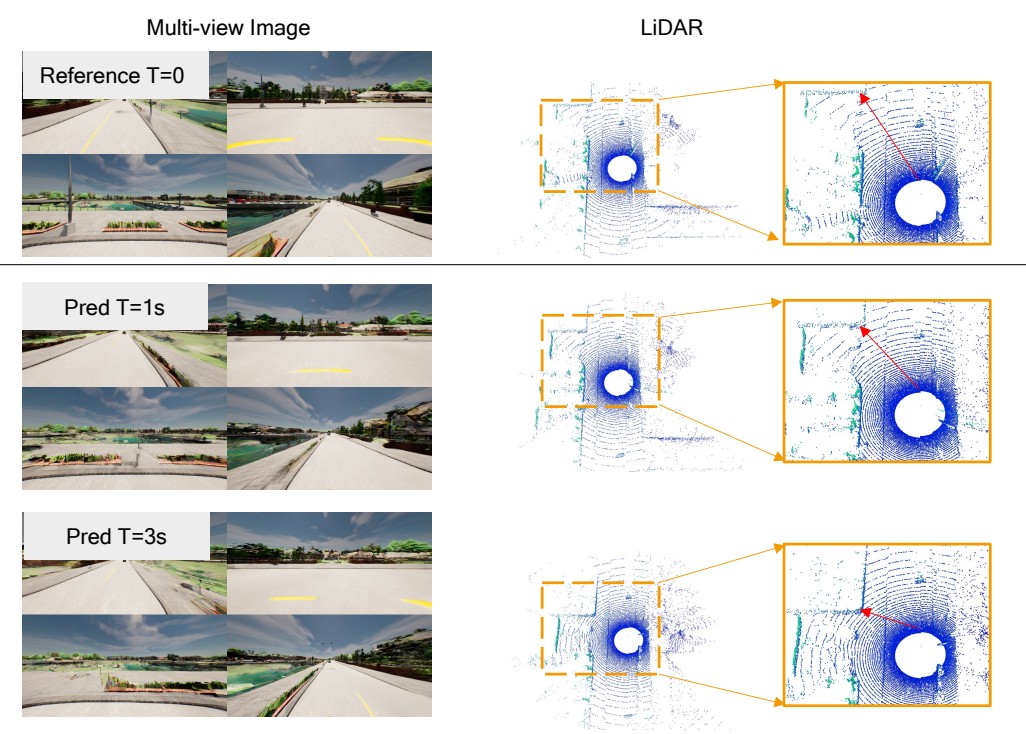

Figure 10: The visualization of LiDAR and future predictions on Carla dataset.

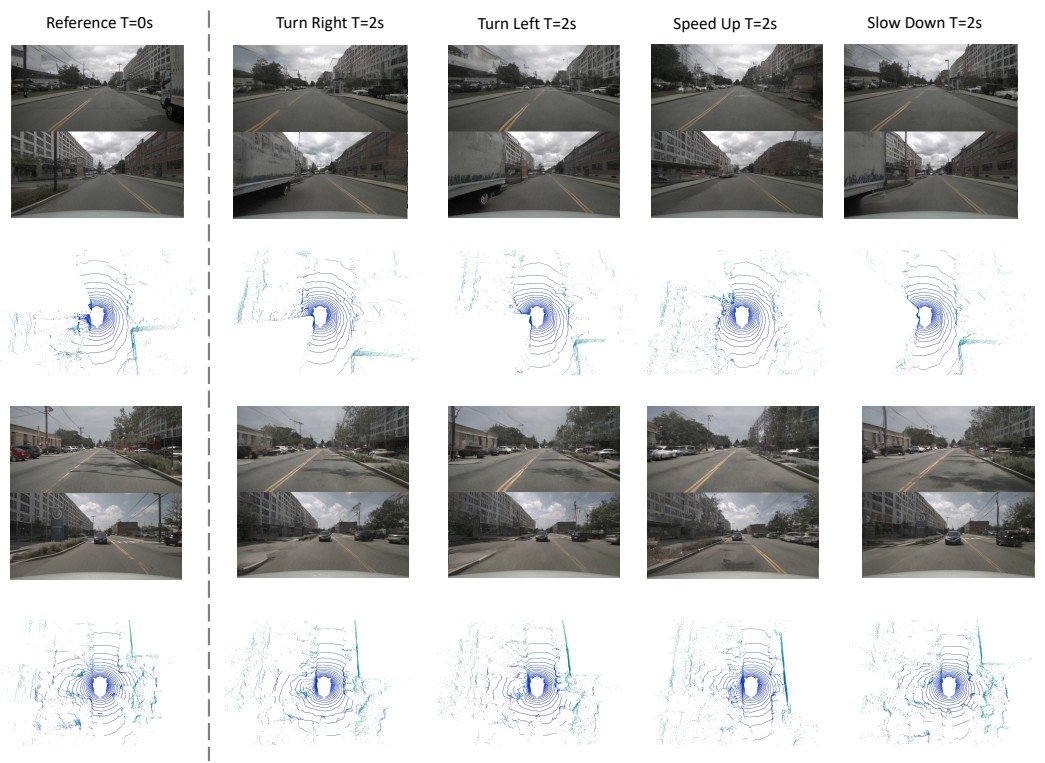

Figure 11: More visual results of controllability.

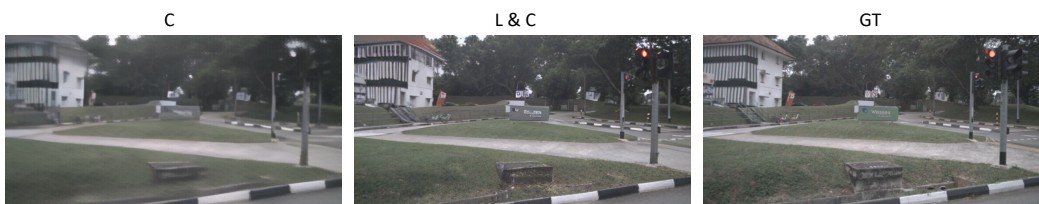

Figure 12: The visualization of C and L & C.

cosine learning rate decay strategy. The multi-task loss function includes a perceptual loss weight of 0.1, a lidar loss weight of 1.0, and an RGB L1 reconstruction loss weight of 1.0. For the GAN training, we employed a warm-up strategy, introducing the GAN loss after 30,000 iterations. The discriminator loss weight was set to 1.0, and the generator loss weight was set to 0.1.

**Details on Upsampling from 2D BEV to 3D Voxel Features.** The dimensional transformation proceeds as follows: $(4, 96, 96)$ $\xrightarrow{\text{Step1: a linear layer}}$ $(256, 96, 96)$ $\xrightarrow{\text{Step2: Swin Blocks and upsampling}}$ $(128, 192, 192)$ $\xrightarrow{\text{Step3: additional Swin Blocks}}$ $(128, 192, 192)$ $\xrightarrow{\text{Step4: a linear layer}}$ $(4096, 192, 192)$ $\xrightarrow{\text{Step5: reshaping}}$ $(16, 64, 384, 384)$. For the upsampling in Step 2, we adopt Patch Expanding, which is commonly used in ViT-based approaches and can be seen as the reverse operation of Patch Merging. The linear layer in Step 4 predicts a local region of shape $(16, 64, r_y, r_x)$, where spatial sizes are adjusted (e.g., $r_y$=2, $r_x$=2), followed by reshaping in Step 5 to the final 3D feature shape.

**Composition of 3D Voxel Features.** Along each ray, we perform uniform sampling, and the depth t of the sampled points is a predefined value, not predicted by the model. The feature $\mathbf{v}_i$ at these sampled points is obtained through linear interpolation, while the blending weight w is predicted from the sampled features $\mathbf{v}_i$ (as described in Equation 1). This is a standard differentiable rendering process.

## C  BROADER IMPACTS

The concept of a world model holds significant relevance and diverse applications within the realm of autonomous driving. It serves as a versatile tool, functioning as a simulator, a generator of long-tail data, and a pre-trained model for subsequent tasks. Our proposed method introduces a multi-modal BEV world model framework, designed to align seamlessly with the multi-sensor configurations inherent in existing autonomous driving models. Consequently, integrating our approach into current autonomous driving methodologies stands to yield substantial benefits.

## D  LIMITATIONS

It is widely acknowledged that inferring diffusion models typically demands around 50 steps to attain denoising results, a process characterized by its sluggishness and computational expense. Regrettably, we encounter similar challenges. As pioneers in the exploration of constructing a multi-modal world model, our primary emphasis lies on the generation quality within driving scenes, prioritizing it over computational overhead. Recognizing the significance of efficiency, we identify the adoption of one-step diffusion as a crucial direction for future improvement in the proposed method. Regarding the quality of the generated imagery, we have noticed that dynamic objects within the images sometimes suffer from blurriness. To address this and further improve their clarity and consistency, a dedicated module specifically tailored for dynamic objects may be necessary in the future.

