# OpenReview forum: "BEVWorld: A Multimodal World Model for Autonomous Driving via Unified BEV Latent Space"
_ICLR.cc/2025/Conference — ICLR 2025 Conference Withdrawn Submission_

### Official Review · Reviewer_gbNQ · 2024-11-03

**Soundness:** 3
**Presentation:** 2
**Contribution:** 2
**Rating:** 6
**Confidence:** 3

**Summary:**

The authors presented BEVWorld, a pipeline that first tokenizes multi-modal observations, and predicts and decodes future image and lidar observations conditioned on action sequences. The approach is to auto-encode sensor inputs into BEV latent space, and predict the latent evolutions through a BEV sequence diffusion model.

**Strengths:**

1. The authors proposed a rather thorough pipeline, involving multimodal sensor inputs, to work well compared to existing appraoches.
2. The multimodal encoder decoder architecture shows reasonably good performance and effectiveness for downstream tasks.

**Weaknesses:**

1. See the questions section.
2. Writing clarity and organization may be improved, such as in the related work section.

**Questions:**

1. The rather complicated pipeline seems to be mostly motivated by a realistic autonomous driving setting. How costly is the pipeline, and would it actually work in deployment? Would the authors argue that the novelty/contribution lies more in its application or methodological insights?
2. The tokenizer decoder's multi-view lidar and image rendering seems to be a complicated (and potentially costly due to ray sampling similar to NeRF's I assume?) Is the motivation solely to maintain consistency between the two modalities? Have the authors thought of other representations such as 3D Gaussian Splatting that could be compatible for both lidar pointcloud and image projection?
3. Around line 332, the authors mention "various choice of input and output modalities," whereas only camera and lidar are mentioned throughout the paper. If "various ... modalities" means more than camera and lidar, it shall be better illustrated.
4. Some of the methods in table 4 and 5 operate under different modalities compared to BEVWorld. Would this still give us a fair comparison? (sorry if I am not super familiar with the other methods that the authors compare against)
5. How does the visual and lidar predictions deal with potential uncertainty in the scene? This does not seem to be mentioned in the paper, but I am curious of the role that uncertainty plays in the application of these models.

---

### Official Review · Reviewer_ozXM · 2024-11-03

**Soundness:** 3
**Presentation:** 2
**Contribution:** 2
**Rating:** 3
**Confidence:** 3

**Summary:**

The authors propose a multimodal encoder that  learns a world model using a stream of multi-view images and point cloud. This novel approach tokenizes multimodal sensor inputs into a unified and compact Bird’s Eye View (BEV) latent space and could be used for environment modeling in downstream autonomous driving tasks.

**Strengths:**

Strengths:

1. The related section is well laid out.
2. The figures are informative and as a reader I’m able to capture the essence of the paper quickly.

**Weaknesses:**

Weekness:

1. Upon googling some related work, I found this paper “Bird's Eye View Based Pretrained World model for Visual Navigation; Lekkala et al.” that actually does planning and control tasks in visual navigation using a dataset of FPV and BEV images. Since you use multi-view imagery and BEV based point cloud, I’d suggest evaluating your representations on some navigation tasks like that of Carla, just similar to how Lekkala et al. performed it, citing the mentioned paper.
2. Currently, the experiment section is quite confusing. I suggest reorganizing the experiment section by first having experiments that demonstrate the efficacy on the downstream tasks. In each of the subsections that deals with downstream tasks, lay out the datasets that were used and the baselines that are used to compare. And in the next section after experiments, you can have an ablations section that consists of the efficacy of the components used.
3. Experiments are very limited. When you mentioned that your proposed method encodes multi-modality information, I was under the impression that these representations are useful in downstream tasks that help in prediction and planning. In regards to planning and motion control, one of my biggest concern is that your method is not compared against other baselines in other tasks. This makes it difficult to accept the benefits offered by the improvements in your method.
4. The presentation of the paper needs some work, especially the captions of the tables. Currently it’s hard to understand what experiments each table is reporting. In Table 5. What is the task that the results are being reported.
5. I understand that one of the proposed mechanisms involve a bigger model that is capable of understanding multi view images and lidar point clouds. I appreciate the authors evaluating on prediction tasks to detect 3D detection.
6. The motion prediction task is not clear. How can you predict motion using the data at the current/previous time steps without considering the actions taken in the next time steps?

**Questions:**

Please elaborately address every point in the above weakness section.

---

### Official Review · Reviewer_JsbB · 2024-11-04

**Soundness:** 3
**Presentation:** 3
**Contribution:** 2
**Rating:** 5
**Confidence:** 4

**Summary:**

The paper proposes BEVWorld, a novel multimodal world model for autonomous driving. By unifying camera and LiDAR data into a BEV latent space, BEVWorld achieves compact and coherent environment modeling, which is used for future scene prediction in a diffusion-based approach. The model exhibits strong performance on future prediction and downstream tasks compared to existing methods, as demonstrated through experiments on the nuScenes and Carla datasets.

**Strengths:**

1. Unified Multimodal Representation: This paper introduces a novel multimodal tokenizer that integrates visual semantics and 3D geometric information into a unified Bird's Eye View (BEV) representation.
2. Latent Diffusion Model: A latent diffusion-based world model is designed to simultaneously generate future multi-view images and point clouds. This model avoids the problem of error accumulation found in autoregressive approaches, and extensive experiments on the nuScenes and Carla datasets demonstrate its superior performance in future predictions using multimodal data.
3. Controllable Generation: The generated video and LiDAR data can be controlled through action conditions, such as commands for turning left, turning right, accelerating, and decelerating. The resulting images and LiDAR data can be adjusted according to these instructions, showcasing the model's potential for controllability.
4. Effectiveness for Downstream Tasks: The effectiveness of the future BEV features predicted by the latent diffusion network for planning tasks is validated through experiments in downstream open-loop planning tasks. Additionally, this paper validates its broad effectiveness across different datasets.

**Weaknesses:**

1. Computational Efficiency: The diffusion process is computationally intensive, which may hinder real-time applicability. The model requires multiple steps to generate denoised outputs, leading to slower inference times.
2. Blurred Dynamic Objects: The paper mentions that dynamic objects sometimes appear blurry in generated images, potentially impacting model reliability in high-speed or densely populated environments.
3. Dependency on Action Tokens: The model's prediction quality may rely heavily on the quality and accuracy of the action tokens, which may vary in complex real-world scenarios.
4.Dependence on the Number of Historical Condition Frames: In the comparison, BEVWorld requires 3 historical frames for a 3-second prediction, while Copilot4D uses 6 frames.
5. Insufficient Novelty: DiffBEV has also demonstrated the potential of combining BEV with diffusion, yet this paper does not reference DiffBEV. Additionally, 3D multimodality is a well-established field, and the paper fails to cite works like BEVFusion.

**Questions:**

1. Why the results of 3D Detection are significantly lower than previous Multi-modal fusion methods: CMT, BEVFusion, and DeepInteraction？
2. The experiments do not clearly demonstrate whether future predictions contribute to improving downstream tasks. For instance, it would be beneficial to directly compare the results of multimodal fusion 3D detection with those of 3D detection that includes future frames.
3. Given the computational demands of the diffusion model, is BEVWorld feasible for real-time applications, especially for high-frequency sensor inputs in autonomous driving?

---

### Official Review · Reviewer_2JVb · 2024-11-04

**Soundness:** 3
**Presentation:** 3
**Contribution:** 2
**Rating:** 5
**Confidence:** 4

**Summary:**

This work presents BEVWorld, a diffusion-based world model operating in bird's-eye-view (BEV) space for autonomous driving.
They pretrain the latent space by reconstructing lidar and camera inputs with their multi-modal tokenizer, and perform next BEV prediction using a latent BEV sequence diffusion model.
Their architecture consists of a BEV encoder (using Swin Transformer and PointPillars), BEV decoder, and a multi-modal rendering network.
After pretraining, they perform downstream training and evaluation on autonomous driving perception tasks (3D detection, motion prediction) on nuScenes and demonstrate video generation capabilities.

**Strengths:**

The idea and motivation are novel as there has yet to be a good BEV pretraining objective for perceptions tasks.

The method shows improvements in standard autonomous driving tasks (3D detection, motion prediction) when used as a pretraining approach.
Results on generation are also strong compared to baselines.

Writing and technical content are written and illustrated clearly in the figures.

**Weaknesses:**

Multiple components of the method appear to build on existing approaches without clear acknowledgment or discussion.
The BEV feature lifting via projection and deformable attention bears strong similarity to BEVFormer's spatial cross-attention.
Similarly, the lidar ray-casting in Sec 3.1 also bears resemblance to some existing methods in neural reconstruction.
It would be helpful to either cite these works or clarify how their approach differs.

There are discrepancies in the video generation metrics compared to prior work.
For instance, their reported FID of 52.6 (Table 4) is substantially worse than the original results in DriveDreamer's (14.9), but the evaluation protocol differences aren't clearly explained.

Baselines for Table 3 only include their own implementations and comparison with specialist models are missing.

**Questions:**

What are the key differences in evaluation protocol compared to DriveDreamer that might explain the large FID discrepancy (52.6 vs 14.9)?

The motivation for using 4-dimensional features in the future prediction task is unclear.
The authors claim this task "requires low-dimension inputs" but don't adequately justify this design choice.
It is quite surprising that the BEV space can be represented with such a bottleneck.

---

### Official Review · Reviewer_YRJC · 2024-11-04

**Soundness:** 1
**Presentation:** 3
**Contribution:** 2
**Rating:** 3
**Confidence:** 4

**Summary:**

The paper presents a new driving world model named BEVWorld. It processes multi-modal sensor inputs (camera and LiDAR) and operates within a shared BEV latent space. This latent space is learned via a multi-modal tokenizer, supervised by image and LiDAR reconstruction. To enable world modeling, the paper introduces a diffusion model to predict future latents conditioned on ego motions and previous BEV latents. The paper also applies the proposed BEVWorld to several downstream tasks, including 3D detection, motion prediction, and planning.

**Strengths:**

S1) The authors conduct downstream applications on both nuScenes and CARLA, covering 3 vital tasks in autonomous driving.

S2) Quantitative evaluation aside, the authors provide extensive qualitative visualizations to demonstrate the fidelity of BEVWorld.

S3) The writing and figures are well-presented, and the implementation details are clearly specified, making the paper easy to follow.

**Weaknesses:**

W1) The motivation of the paper appears confusing. The key idea is to develop a multi-modal world model, but the connection between this idea and the experiments is unclear. Specifically, the downstream applications in 3D detection and motion prediction only demonstrate the effectiveness of tokenizer pretraining, rather than future prediction. On the other hand, the planning experiment seems irrelevant to the functionality of a world model. Since a world model serves as a transition function controlled by actions, it seems odd to totally drop action conditions and use the model as a planner. Note that the proposed BEVWorld does not incorporate a classifier-free guidance strategy during training, meaning the model must take actions as inputs and the planning situation is never handled by the model. Thus, I don’t think it is a proper way to apply the world model.

W2) A standard evaluation of all planning metrics is missing. The submitted paper only reports the L2 3s metric. In addition, the abstract only mentions 3D detection and motion prediction. Is that because the planning part is too weak to emphasize?

W3) The evaluation of action controllability is inadequate. It would be more comprehensive and convincing with qualitative and quantitative comparisons to previous methods.

W4) Since one of the main novelties is the multi-modal design, it would be beneficial to demonstrate its necessity in terms of downstream performance.

**Questions:**

Q1) The architecture design and loss function remind me of ViDAR [CVPR 2024]. Which parts are similar or different is not clearly explained. Could you elaborate?

Q2) The usefulness of the classifier-free guidance strategy has been verified by many world models in autonomous driving (e.g., the results of Copilot4D in Table 5). It should be effective in improving prediction fidelity and action controllability, aligning with the objectives of a world model. Did you try to train a CFG version of your model? If yes, how is its performance? If no, why not?

Q3) Is it possible to add more baselines when comparing the effectiveness of the pretrained tokenizer in Table 3? From the current form, it is somewhat hard to conclude the superiority of the proposed multi-modal pretraining method.

---

### Note · Authors · 2024-11-22

I have read and agree with the venue's withdrawal policy on behalf of myself and my co-authors.